# Stacked antiaromatic porphyrins

Ryo Nozawa[1], Hiroko Tanaka[1], Won-Young Cha[2], Yongseok Hong[2], Ichiro Hisaki[3], Soji Shimizu[4], Ji-Young Shin[1], Tim Kowalczyk[5], Stephan Irle[6], Dongho Kim[2] & Hiroshi Shinokubo[1]

Aromaticity is a key concept in organic chemistry. Even though this concept has already been theoretically extrapolated to three dimensions, it usually still remains restricted to planar molecules in organic chemistry textbooks. Stacking of antiaromatic π-systems has been proposed to induce three-dimensional aromaticity as a result of strong frontier orbital interactions. However, experimental evidence to support this prediction still remains elusive so far. Here we report that close stacking of antiaromatic porphyrins diminishes their inherent antiaromaticity in the solid state as well as in solution. The antiaromatic stacking furthermore allows a delocalization of the π-electrons, which enhances the two-photon absorption cross-section values of the antiaromatic porphyrins. This feature enables the dynamic switching of the non-linear optical properties by controlling the arrangement of antiaromatic π-systems on the basis of intermolecular orbital interactions.

[1] Department of Applied Chemistry, Graduate School of Engineering, Nagoya University, Nagoya 464-8603, Japan. [2] Department of Chemistry, Yonsei University, Seoul 120-749, Korea. [3] Department of Material and Life Science, Graduate School of Engineering, Osaka University, Osaka 565-0871, Japan. [4] Department of Applied Chemistry, Graduate School of Engineering, Kyushu University, Fukuoka 819-0395, Japan. [5] Department of Chemistry, Advanced Materials Science and Engineering Center, and Institute for Energy Studies, Western Washington University, Bellingham, Washington 98225, USA. [6] Department of Chemistry, Graduate School of Science, Nagoya University, Nagoya 464-8602, Japan. Correspondence and requests for materials should be addressed to T.K. (email: Tim.Kowalczyk@wwu.edu) or to D.K. (email: dongho@yonsei.ac.kr) or to H.S. (email: hshino@apchem.nagoya-u.ac.jp).

The parallel orientation of two or more planar π-conjugated molecules is usually referred to as π–π-stacking[1-4], which often controls the structures of supramolecules and liquid crystalline materials. It is also important in biological systems, where it controls in many cases the structure and functionality of DNA and proteins (tertiary structure)[5]. Stacked aromatic π-systems are also key components in optoelectronic organic devices, as the close contact of π-systems with a large orbital overlap offers an effective conduction pathway for charge carriers, which affords high-performance conducting materials[6].

In the solid state, large planar π-conjugated molecules often adopt π–π stacked structures with interplanar distances that typically range from 3.4 to 3.6 Å, that is, the sum of van der Waals radii for sp²-hybridized carbon atoms. However, a complete overlap of two π-systems (face-centred stacking) would be difficult on account of the resulting severe electrostatic repulsion between the π-electrons. Consequently, a slipped stacked structure (offset stacking) is more commonly observed, where dispersion forces between π-electrons and peripheral protons dominate the attractive interaction between two π-systems[7]. However, the orbital interactions between the π-systems are substantially reduced in the offset stacking. To achieve more effective intermolecular electronic communication via a closer stacking of π-conjugated molecules, a new approach is required.

The stacking of antiaromatic compounds represents one promising strategy. Corminboeuf et al.[8] have proposed that stacking of two antiaromatic systems in methano-bridged superphanes such as cyclobutadiene dimer **1** (Fig. 1) can eliminate their antiaromaticity due to the resulting three-dimensional aromaticity, which results from mutual interactions between frontier orbitals of each π-system. This intriguing proposal was further supported by a theoretical study of Bean and Fowler[9] on the ring current effect in antiaromatic cyclophanes with a very short interplanar distance between the two π-systems (∼2.2 Å). In such systems, intermolecular electronic delocalization should be significantly increased. However, experimental support to substantiate this prediction still remains elusive on account of the synthetic difficulties associated with constructing superphanes from unstable antiaromatic compounds[10].

We have recently developed an efficient synthesis of stable antiaromatic porphyrins such as dimesitylnorcorrole Ni(II) (**3a**), which can be obtained on a gram scale[11]. Norcorrole is a ring-contracted porphyrin that contains two *meso*-carbons fewer than a regular porphyrin[12]. According to the Hückel rule, norcorrole Ni(II) complexes should be characterized as distinctly antiaromatic, given their 16 π-electronic circuit and the planar structure. Subsequently, we attempted the synthesis of norcorroles with less bulky peripheral substituents in order to facilitate π–π stacking in the solid state. We discovered that diphenylnorcorrole Ni(II) (**3b**) adopts a triple-decker π–π stacked structure in the solid state with a very short interplanar distance (3.149 Å). Most importantly, a significant reduction of

antiaromaticity of the norcorrole skeleton was observed for **3b** in the solid state. Following that discovery, we constructed the tethered norcorrole dimers **5a** and **5b** to confirm the substantially diminished antiaromaticity of the stacked antiaromatic systems in solution. These stacked norcorroles represent the experimental evidence for the emergence of aromaticity in antiaromatic π-systems upon stacking.

## Results

**Synthesis and properties of diphenylnorcorrole Ni(II).** The synthetic route from dibromodipyrrin Ni(II) complex **2b** to diphenylnorcorrole Ni(II) (**3b**) is shown in Fig. 2a. A Ni(0)-mediated intramolecular cyclization of **2b** afforded **3b** in 28% yield. Compound **3b** was characterized by multinuclear NMR spectroscopy and high-resolution mass spectrometric analyses. The $^1$H NMR spectrum of **3b** displayed two sets of pyrrole protons in the far upfield region ($\delta = 1.7$–2.2 p.p.m.), owing to a strong paratropic ring current effect, which clearly demonstrates distinct antiaromaticity for **3b** in solution. Even though **3b** is relatively stable in the solid state, slow oxidation was observed in solution; no such oxidation was observed for dimesitylnorcorrole **3a**.

A single-crystal X-ray diffraction analysis unambiguously confirmed that **3b** adopts a triple-decker stacking structure in the solid state (Fig. 2b), which is in stark contrast to the herringbone packing structure that **3a** adopts. In the triple-decker structure, the Ni atom of the central molecule lies on a centre of symmetry. The distance between the two nickel centres is 2.998 Å, while the distance between the two mean planes of the four nitrogen atoms of each macrocycle is 3.149 Å (Supplementary Fig. 13), which is much shorter than typical π–π stacking distances for aromatic compounds (3.4–3.6 Å)[1,6]. The central norcorrole molecule is relatively planar, while the two outer ones are distorted into a bowl-shaped conformation. As shown in Fig. 2c, both outer molecules are perfectly eclipsed with respect to each other, whereas the central macrocycle is offset by 72°.

The bond length alternation (BLA) in cyclic compounds is a good indicator of aromaticity, as it allows an evaluation of the degree of effective π-electron delocalization. Interestingly, the BLA in triple-decker norcorrole **3b** is significantly smaller than that in **3a**, which does not adopt any π–π stacking in the crystal. For example, the C–C bond lengths around the *meso*-carbon atom are 1.447 and 1.403 Å in **3a**, which is a characteristic feature of antiaromatic porphyrinoids. In contrast, the C–C bond lengths around the *meso*-carbon atom in **3b** are 1.441 and 1.422 Å (central molecule), as well as 1.438 and 1.419 Å (outer molecules). The BLA is often quantified on the basis of the harmonic oscillator model of aromaticity (HOMA) values, which are close to 1 in aromatic molecules[13]. For antiaromatic dimesitylnorcorrole **3a**, a HOMA value of 0.45 was obtained, while for stacked norcorrole **3b** values of 0.58 (outer molecules) and 0.56 (central molecule) were observed (Supplementary

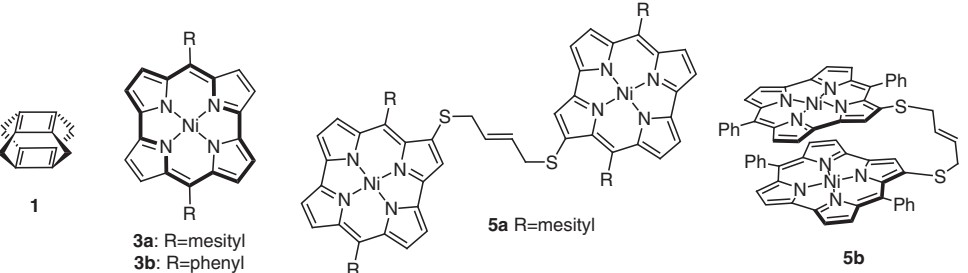

**Figure 1 | Structures of antiaromatic superphane 1 and norcorroles.** The bold lines in **3a** and **3b** indicate one of their 16 π-electronic circuits.

Fig. 15). These values are quite substantial, considering that aromatic tetramesitylporphyrin exhibits a HOMA value of 0.77. Accordingly, the attenuated BLA in **3b** strongly supports an increase in aromaticity through stacking of antiaromatic π-systems, as predicted by Corminboeuf et al.[8] and Bean and Fowler[9].

**Synthesis and properties of a stacked norcorrole dimer.** To elucidate the properties of the stacked norcorrole Ni(II) complex in solution, we synthesized and investigated tethered dimer **5b**. The synthesis of **5b** started with the introduction of allylthiol to **3b** to provide allylthionorcorrole **4b** (Fig. 3a)[14]. Subsequently, **4b** was dimerized using a second-generation Hoveyda–Grubbs complex to afford **5b** in 58% yield[15]. For comparison with non-stacked analogues, mesityl-substituted dimer **5a** was prepared in a similar manner. A single-crystal X-ray diffraction analysis of **5b**

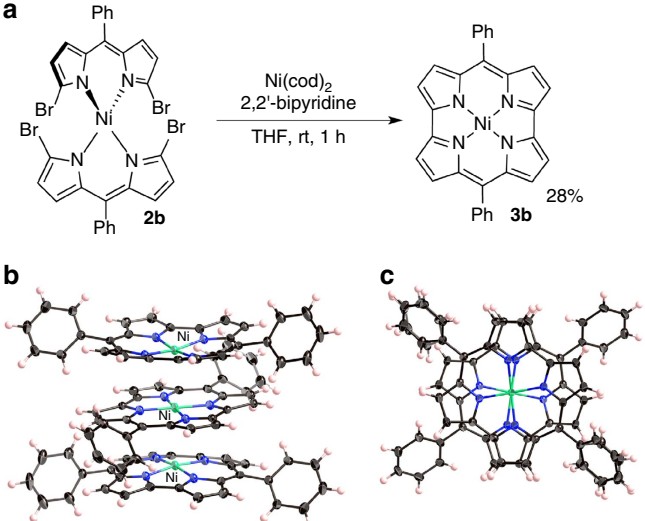

**Figure 2 | Synthesis and solid-state structure of 3b.** (**a**) Synthetic route to **3b**, (**b**) oblique view of **3b** and (**c**) top view of **3b** (atomic displacement parameters set at 50% probability).

revealed a closely stacked structure with very short interplanar (3.050 Å) and Ni–Ni distances (2.849 Å) (Fig. 3b and Supplementary Fig. 14). For the norcorrole cores in **5b**, HOMA values of 0.56 and 0.54 were observed, which corroborates the significant decrease of the inherent antiaromaticity of the norcorrole unit in the solid state (Supplementary Fig. 15).

With stacked dimer **5b** in hand, we went on to explore its solution properties. Interestingly, the [1]H NMR spectrum of **5b** exhibited considerably down-field shifted resonances for the pyrrole protons relative to those of monomeric norcorroles **3** or non-stacked dimer **5a**. The pyrrole protons of **5b** were observed at 3.5–4.7 p.p.m., while those of **3a** appeared at 1.57 and 1.47 p.p.m. Upon lowering the temperature, the pyrrole protons of **5b** experienced a further down-field shift in the [1]H NMR spectrum (Supplementary Fig. 11). This change indicates the presence of an equilibrium between stacked and non-stacked conformers of **5b**, in which the former predominates at lower temperatures. The chemical shift change was analysed using the van't Hoff equation (Supplementary Fig. 12). The extrapolated chemical shifts for entirely stacked **5b** should be expected at 3.8–5.1 p.p.m. Such down-field shifts were not observed for non-stacked dimer **5a**.

The drastic down-field shifts of the proton signals in stacked **5b** could either be ascribed to the weakened antiaromaticity of the individual norcorrole unit or to the paratropic ring current effect of the other norcorrole unit. To clarify the origin of the down-field shift, the ring currents in **3a** and **5b** were visualized using anisotropy of the induced current density (ACID)[16] plots (Supplementary Fig. 18). These clearly demonstrated an attenuated current density in **5b** and thus refute the latter possibility. The nucleus-independent chemical shift (NICS)[17] (Fig. 3c) as well as two-dimensional NICS plot (Supplementary Fig. 19) of **5b** revealed a considerably smaller magnetic effect of each norcorrole macrocycle compared with monomer **3a**, thus supporting the weakened antiaromaticity in **5b**. Taking also the bond length equalization observed in the crystal structure of **5b** into account, we thus conclude that a close stacking of two antiaromatic π-systems results in a substantial decrease of their individual antiaromaticity.

Generally, aromaticity in molecules should provide energetic stabilization by electronic delocalization. Indeed, we found that

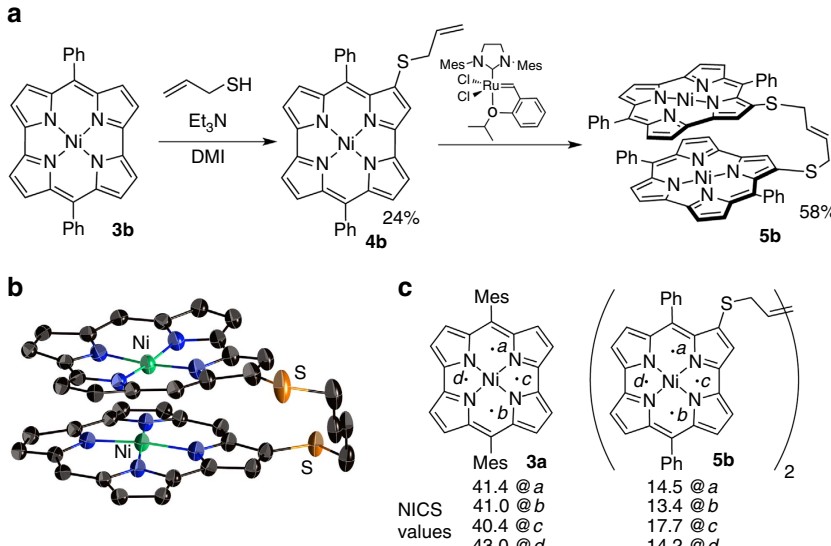

**Figure 3 | Synthesis and aromaticity of stacked norcorrole dimer 5b.** (**a**) Synthetic route to **5b** (DMI = 1,3-dimethyl-2-imidazolidinone), (**b**) molecular structure of **5b** (atomic displacement parameters set at 50% probability; phenyl groups and hydrogen atoms omitted for clarity) and (**c**) NICS values of **3a** and **5b** at several points (mean values of two norcorrole units).

NICS values

**3a**
41.4 @a
41.0 @b
40.4 @c
43.0 @d

**5b**
14.5 @a
13.4 @b
17.7 @c
14.2 @d

the dimer **5b** is certainly more stable than monomer **4b** (Fig. 4). While heating **4b** in toluene-$d_6$ to 80 °C under an atmosphere of air resulted in virtually quantitative decomposition (97%), stacked dimer **5b** could be recovered in 92% under identical conditions, thus demonstrating enhanced chemical stability relative to **5b**.

Antiaromatic porphyrinoids typically exhibit weak and broad absorption bands in the near infrared region ($>800$ nm). This is also the case for monomer **4b** (Fig. 5a, bottom). In contrast, the absorption spectrum of dimer **5b** does not display this characteristic feature of antiaromatic porphyrinoids (Fig. 5b, bottom). Interestingly, **5b** showed a distinct absorption band at $\sim 860$ nm. This low-energy absorption band increased at 77 K, which supports a shift of the equilibrium between the stacked and the non-stacked conformer towards the former at low temperatures (Supplementary Fig. 16). In contrast, **5a**, in which $\pi$–$\pi$ stacking is not possible on account of the bulky mesityl substituents, did not exhibit any such absorption band but only weak, broad absorption bands in the near infrared region, similar to the monomers. Time-dependent density functional theory calculations revealed that this peak should contain significant contributions of the transition from the highest occupied molecular orbital (HOMO) to the lowest unoccupied molecular orbital (LUMO) and from the HOMO$-1$ to the LUMO$+1$; the HOMO and HOMO$-1$, as well as the LUMO and LUMO$+1$, are virtually degenerate (Supplementary Fig. 20). Such degenerate frontier orbitals are typical for aromatic porphyrins. In addition, the molecular absorption extinction coefficient of **5b** is less than half of those of **3a** and **3b**. This feature is intriguing, as the dimer of a chromophore should usually exhibit an almost doubled molecular absorption extinction coefficient in the absence of inter-chromophore interaction. This result thus indicates that the electronic structure of **5b** should be significantly different from that of the monomer on account of the substantial spatial electronic interactions between the two norcorrole $\pi$-systems.

In order to obtain further experimental insight into the electronic structure, we measured the magnetic circular dichroism (MCD) spectrum of **5b** (Fig. 5b, top). The MCD spectrum of **5b** showed a weak but nevertheless distinct Faraday B term at $\sim 850$ nm. Based on the perimeter model[18–21], the lowest-energy band of norcorrole monomers should be forbidden owing to the intrashell nature. Accordingly, the absorption and MCD band observed for **5b** at $\sim 850$ nm should be ascribed to transitions between the molecular orbitals of the stacked two norcorrole units. It is reasonable to assume that the overlap between these orbitals should increase the probability of the otherwise forbidden HOMO–LUMO transition in the norcorrole monomers.

**Two-photon absorption (TPA) properties of the stacked dimer**. In porphyrinoids, the TPA cross-section values usually correlate with the degree of aromaticity. The TPA process is a third-order nonlinear optical phenomenon and its occurrence is largely related to $\pi$-electron delocalization, that is, TPA cross-section values can be employed to evaluate the aromaticity of porphyrinoids[22–24]. In general, aromatic porphyrinoids exhibit substantially larger TPA cross-section values than their antiaromatic counterparts, mostly owing to a more effective $\pi$-electron delocalization. The TPA cross-section value of stacked dimer **5b** in toluene (1,000 GM at 1,700 nm) is significantly higher than those of monomer **3a** (180 GM at 1,700 nm) and non-stacked dimer **5a** (220 GM at 1,800 nm) (Supplementary Fig. 17). This result suggests effective $\pi$-electron delocalization through three-dimensionally extended $\pi$-orbital overlap, which is consistent with HOMA and NICS values, as well as with the ACID analysis. These results also demonstrate that nonlinear optical properties of antiaromatic $\pi$-systems can be dynamically tuned by controlling their supramolecular alignment.

**Electronic origins of the stabilization in the stacked dimer**. We observed that the antiaromatic $\pi$-system of the norcorroles easily forms close face-to-face stacking structures in order to diminish their antiaromatic nature. This phenomenon is not generally observed in aromatic compounds, for which close face-to-face $\pi$-stacking typically leads to large repulsive forces between the stacked $\pi$-electrons. To explain the difference in intermolecular interactions underlying this contrasting feature, we carried out

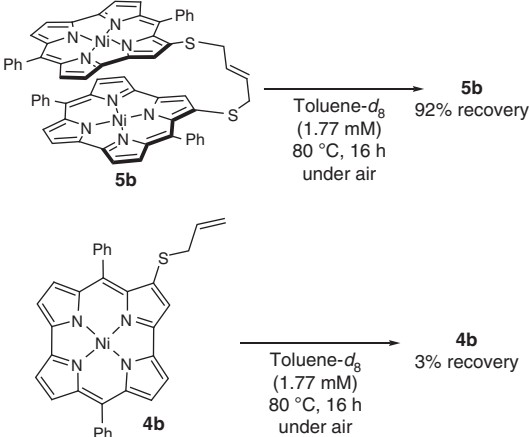

**Figure 4 | Stability of 5b and 4b.** Thermal decomposition experiments of **4b** and **5b** at 80 °C under air.

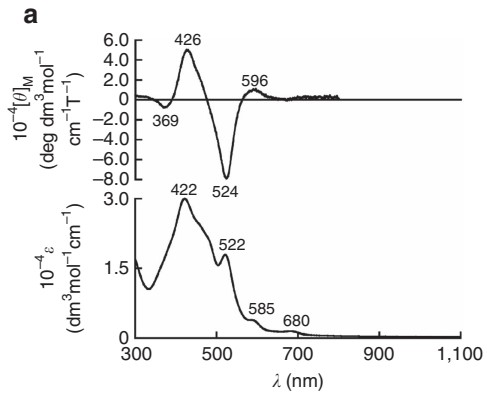
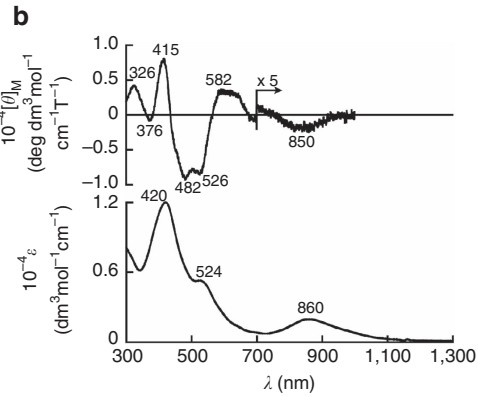

**Figure 5 | Optical properties of 4b and 5b.** (**a**) Absorption (bottom) and MCD (top) spectra of norcorrole monomer **4b** in CH$_2$Cl$_2$ and (**b**) absorption (bottom) and MCD (top) spectra of norcorrole dimer **5b** in CH$_2$Cl$_2$.

natural bond orbital (NBO) analyses[25] on antiaromatic Ni(II) norcorrole dimer **3b** and on a comparable aromatic Ni(II) porphyrin dimer. In **3b**, much stronger intermolecular orbital interactions were observed than in the corresponding stacked porphyrin dimer. On the basis of an NBO analysis, we thus conclude that the chemical origin of the stacking should be attributed to significantly stronger donor–acceptor interactions between the occupied orbitals on the carbon atoms of one monomer and the $\pi^*$ orbitals of the other monomer (Supplementary Table 2). Furthermore, a Wiberg bond order[26] of $<0.2$ was observed for the Ni–Ni interaction in **3b** at a Ni–Ni separation of 3.0 Å (Supplementary Fig. 21). These results indicate that the intermolecular orbital interactions between the norcorrole units outweigh the Ni–Ni bonding in the stacked dimer. Consequently, our results on the increased aromaticity of norcorroles upon dimerization are consistent with previous theoretical predictions for antiaromatic superphanes.

## Discussion

In summary, we have presented an experimental example of closely stacked antiaromatic $\pi$-conjugated systems using norcorrole Ni(II) complexes. In the stacked norcorroles, significantly diminished antiaromaticity was observed, as well as the emergence of aromatic features such as enhanced stability, bond length equalization and effective electronic delocalization. As stacked antiaromatic compounds allow an exceptionally close alignment of $\pi$-conjugated systems and accordingly large intermolecular orbital interactions, they constitute a design principle for materials with potential applications in organic electronics. Furthermore, this study demonstrates the possibility to tune the non-linear optical properties of such antiaromatic $\pi$-systems dynamically by controlling their spatial arrangement through supramolecular interactions. Although antiaromatic compounds currently still lack practical applications, this study should provide the conceptual basis for a multitude of applications in nonlinear optical materials.

## Methods

**Materials and characterization.** [1]H NMR (500 MHz) and [13]C NMR (126 MHz) spectra of the compounds were recorded on a Bruker AVANCE III HD spectrometer. Chemical shifts were reported as the delta scale in p.p.m. relative to CDCl$_3$ ($\delta = 7.26$ p.p.m.) for [1]H NMR and CDCl$_3$ ($\delta = 77.16$ p.p.m.) for [13]C NMR. [1]H and [13]C NMR spectra are provided for compounds; see Supplementary Figs 1–10. Ultraviolet/Vis/near-infrared spectroscopy (NIR) absorption spectra were recorded on a Shimadzu UV-2550 or JASCO V670 spectrometer. Mass spectra were recorded on a Bruker microTOF using electrospray ionization (ESI)-time-of-flight method for acetonitrile solutions. Unless otherwise noted, materials obtained from commercial suppliers were used without further purification.

**Synthesis of compounds.** *Meso*-phenyl-$\alpha,\alpha'$-dibromodipyrrin Ni(II) complex **2b**. To a mixture of *meso*-phenyl-$\alpha,\alpha'$-dibromodipyrrin (2.43 g, 6.43 mmol) and Ni(OAc)$_2$•4H$_2$O (808 mg, 3.23 mmol), CH2Cl2 (58 ml) and methanol (20 ml) were added. After 1 h of stirring at room temperature, the mixture was evaporated under reduced pressure and the solid residue was recrystallized from CH$_2$Cl$_2$/MeOH to afford *meso*-phenyl-$\alpha,\alpha'$-dibromodipyrrin Ni(II) complex **2b** as a green solid (2.54 g, 3.15 mmol, 98%). The NMR spectra of this compound could not be observed owing to its paramagnetic nature. High-resolution mass spectrometry (HR-MS) ESI-MS: $m/z = 809.7650$, calcd for $(C_{30}H_{18}Br_4N_4Ni)^+ = 809.7613$ $[(M)^+]$.

*Ni(II) meso-diphenylnorcorrole* **3b**. To a mixture of *meso*-phenyl-$\alpha,\alpha'$-dibromodipyrrin Ni(II) complex (**2b**, 327.8 mg, 0.40 mmol), Ni(cod)$_2$ (277.52 mg, 10.0 mmol), and 2,2′-bipyridine (155.8 mg, 10.0 mmol), dry tetrahydrofuran (22.5 ml) was added under argon atmosphere. The solution was stirred at room temperature for 3 h in a glovebox. The solution was passed through alumina pad and evaporated under reduced pressure to provide solid residue. The residue was purified by silica gel column chromatography (hexane/CH$_2$Cl$_2 = 3/1$ as an eluent) to afford Ni(II) *meso*-diphenylnorcorrole **3b** as a green solid (54.13 mg, 0.11 mmol, 28%). [1]H NMR (500 MHz, CDCl$_3$, 1.3 mM): $\delta$ 6.89 (t, 2H, $J = 7.5$ Hz, Ph), 6.67 (t, 4H, $J = 8.0$ Hz, Ph), 5.98 (dd, 4H, $J = 8.0$, $J = 1.0$ Hz, Ph), 2.14 (d, 4H, $J = 4.5$ Hz, $\beta$-H), 1.78 (d, 4H, $J = 4.5$ Hz, $\beta$-H); [13]C NMR (126 MHz, CDCl$_3$, 1.3 mM): $\delta$ 168.08, 158.93, 147.60, 131.98, 131.16, 131.02, 127.89, 120.32, 114.79; Ultraviolet-Vis-NIR (CH$_2$Cl$_2$): $\lambda_{max}$ ($\varepsilon$(M$^{-1}$ cm$^{-1}$)) 431 (46,000), 525 (17,000) nm; HR-MS (ESI-MS): $m/z = 492.0855$, calcd for $(C_{30}H_{18}N_4Ni)^+ = 492.0879$ $[(M)^+]$.

*Synthesis of* **4a**. To a mixture of Ni(II) *meso*-dimesitylnorcorrole **3a** (22.92 mg, 0.04 mmol) and dry 1,3-dimethyl-2-imidazolidinone (2.0 ml) was added. Then allylthiol (16.6 µl, 0.2 mmol) under nitrogen atmosphere, triethylamine (84.0 µl, 0.6 mmol) was added to the solution slowly. The solution was stirred at room temperature for 6 h. Water and EtOAc were then added. The organic layer was separated, washed with brine and evaporated under reduced pressure to leave a solid residue. The residue was purified by silica gel column chromatography (hexane/CH$_2$Cl$_2 = 3/1$ as an eluent) to afford 3-allylthiodimesitylnorcorrole **4a** as a green solid (5.99 mg, 9.2 µmol, 23%). [1]H NMR (500 MHz, CDCl$_3$): $\delta$ 6.31 (s, 2H, Mes), 6.30 (s, 2H, Mes), 5.05–5.01 (m, 1H, vinyl), 4.79 (dd, 1H, $J = 10$ Hz, $J = 1.0$ Hz, vinyl), 4.58 (dd, 1H, $J = 17$ Hz, $J = 1.0$ Hz, vinyl), 2.84 (s, 6H, *ortho*-Me), 2.83 (s, 6H, *ortho*-Me), 2.15 (d, 2H, $J = 6.0$ Hz, CH$_2$), 2.13 (d, 1H, $J = 5.0$ Hz, $\beta$-H), 1.97 (d, 1H, $J = 4.5$ Hz, $\beta$-H), 1.89 (d, 1H, $J = 4.0$ Hz, $\beta$-H), 1.87–1.86 (m, 7H, *para*-Me, $\beta$-H), 1.78–1.76 (m, 2H, $\beta$-H), 1.73 (s, 1H, $\beta$-H); [13]C NMR (126 MHz, CDCl$_3$): $\delta$ 172.64, 171.38, 161.83, 159.72, 157.24, 153.97, 148.64, 147.67, 147.43, 147.16, 145.03, 137.52, 136.95, 133.85, 133.79, 132.69, 131.90, 128.30, 128.15, 126.36, 126.19, 125.54, 124.43, 117.80, 116.51. 113.94, 113.27, 109.60, 34.99, 20.89, 20.72, 17.90, 17.79; Ultraviolet-Vis-NIR (CH$_2$Cl$_2$): $\lambda_{max}$ ($\varepsilon$(M$^{-1}$ cm$^{-1}$)) 428 (28,000), 515 (24,000) nm; HR-MS (ESI-MS): $m/z = 648.1849$, calcd for $(C_{39}H_{34}N_4NiS)^+ = 648.1852$ $[(M)^+]$.

*Synthesis of* **5a**. To a mixture of 3-allylthiodimesitylnorcorrole **4a** (7.40 mg, 11.4 µmol) and second-generation Hoveyda–Grubbs catalyst (7.25 mg, 11.6 µmol), dry dichloromethane (1.0 ml) was added under argon atmosphere in a glovebox. The solution was stirred at room temperature for 4 h. The reaction mixture was filtered through a pad of Celite and concentrated. The residue was purified by silica gel column chromatography (hexane/CH$_2$Cl$_2 = 3/1$ as an eluent) to afford dimesitylnorcorrole dimer **5a** as a green solid (2.25 mg, 1.77 µmol, 31%). [1]H NMR (500 MHz, CDCl$_3$): $\delta$ 6.50 (s, 4H, Mes), 6.31 (s, 4H, Mes), 4.20–4.19 (br, 2H, vinyl), 3.07 (s, 12H, *ortho*-Me), 2.92 (s, 12H, *ortho*-Me), 2.56 (d, 2H, $J = 4.0$ Hz, $\beta$-H), 2.20 (d, 2H, $J = 4.5$ Hz, $\beta$-H), 2.12 (d, 2H, $J = 4.5$ Hz, $\beta$-H), 2.08 (s, 6H, *para*-Me), 1.94–1.93 (m, 6H, CH$_2$, $\beta$-H), 1.91 (d, 2H, $J = 4.0$ Hz, $\beta$-H), 1.86 (d, 2H, $J = 4.5$ Hz, $\beta$-H), 1.85 (s, 6H, *para*-Me), 1.25 (s, 2H, $\beta$-H); [13]C NMR (126 MHz, CDCl$_3$): $\delta$ 172.40, 171.18, 162.49, 160.23, 157.58, 154.22, 147.76, 147.49, 144.98, 137.68, 136.94, 133.99, 133.86, 132.55, 128.44, 128.17, 126.89, 126.43, 126.34, 126.22, 124.82, 116.39, 114.04, 109.45, 33.42, 21.21, 20.71, 18.01, 17.97. (Three sp$^2$-carbon signals were not observed owing to the overlap); Ultraviolet-Vis-NIR (CH$_2$Cl$_2$): $\lambda_{max}$ ($\varepsilon$(M$^{-1}$ cm$^{-1}$)) 428 (43,000) and 516 (36,000) nm; HR-MS (ESI-MS): $m/z = 1268.3417$, calcd for $(C_{76}H_{64}N_8Ni_2)^+ = 1268.3397$ $[(M)^+]$.

*Synthesis of* **4b**. To a mixture of Ni(II) *meso*-diphenylnorcorrole **3b** (9.9 mg, 0.02 mmol) and dry 1,3-dimethyl-2-imidazolidinone (2.0 ml) under nitrogen atmosphere, triethylamine (14.0 µl, 0.2 mmol) was added. The solution was degassed through freeze–pump–thaw three times. Allylthiol (8.3 µl, 0.2 mmol) was added to the solution. The solution was stirred at room temperature for 3 h. Water and EtOAc were then added. The organic layer was separated, washed with brine quickly and evaporated under reduced pressure to leave a solid residue. The residue was purified by silica gel column chromatography (hexane/CH$_2$Cl$_2 = 3/1$ as an eluent) to afford 3-allylthiodiphenylnorcorrole **4b** as a red-black solid (2.7 mg, 4.8 µmol, 24%). [1]H NMR (500 MHz, CDCl$_3$): $\delta$ 6.93–6.83 (m, 4H, Ph), 6.72 (t, 2H, $J = 8.0$ Hz, Ph), 6.21 (d, 2H, $J = 8.0$ Hz, Ph), 6.07–6.05 (d, 2H, $J = 8.0$ Hz, Ph), 5.13–5.06 (m, 1H, vinyl), 4.82 (dd, 1H, $J = 10$ Hz, $J = 1.0$ Hz, vinyl), 4.65 (dd, 1H, $J = 17$ Hz, $J = 1.0$ Hz, vinyl), 2.60 (d, 1H, $J = 4.5$ Hz, $\beta$-H), 2.35 (d, 1H, $J = 4.5$ Hz, $\beta$-H), 2.35 (d, 2H, $J = 4.0$ Hz, CH$_2$), 2.19 (d, 1H, $J = 4.5$ Hz, $\beta$-H), 2.12 (d, 1H, $J = 4.0$ Hz, $\beta$-H), 2.09–2.07 (m, 2H, $\beta$-H), 2.05 (s, 1H, $\beta$-H); [13]C NMR (126 MHz, CDCl$_3$): $\delta$ 171.13, 169.81, 162.41, 161.18, 157.69, 154.75, 148.56, 147.62, 147.18, 146.92, 144.45, 133.47, 131.92, 131.75, 131.50, 130.73, 130.30, 129.27, 128.02, 127.92, 126.84, 123.60, 121.09, 118.07, 116.20, 114.24, 113.72, 111.65, 36.06; Ultraviolet-Vis-NIR (CH$_2$Cl$_2$): $\lambda_{max}$ ($\varepsilon$(M$^{-1}$ cm$^{-1}$)) 422 (30 000) and 519 (19,000) nm; HR-MS (ESI-MS): $m/z = 564.0866$, calcd for $(C_{33}H_{22}N_4NiS)^+ = 564.0913$ $[(M)^+]$.

*Synthesis of* **5b**. A Schlenk tube containing 3-allylthiodiphenylnorcorrole **4b** (3.46 mg, 6.14 µmol) and second-generation Hoveyda–Grubbs catalyst (3.97 mg, 6.14 µmol) was filled with argon before dry dichloromethane (0.4 ml) was added in a glovebox. The solution was stirred at room temperature for 15 min before the reaction mixture was purified by column chromatography on silica gel (eluent: hexane/CH$_2$Cl$_2 = 3/1$, v/v) to afford bridged diphenylnorcorrole dimer **5b** as a black solid (1.97 mg, 3.58 µmol, 58%). [1]H NMR (500 MHz, CDCl$_3$): $\delta$ 7.08–7.03 (m, 8H, Ph), 6.89–6.92 (t, 4H, $J = 7.5$ Hz, Ph), 6.83 (br, 8H, Ph), 4.65 (d, 2H, $J = 4.5$ Hz, $\beta$-H), 4.50 (d, 2H, $J = 4.0$ Hz, $\beta$-H), 4.42 (d, 2H, $J = 4.0$ Hz, $\beta$-H), 4.00–4.02 (m, 2H, vinyl), 3.99 (d, 2H, $J = 4.5$ Hz, $\beta$-H), 3.88 (d, 2H, $J = 4.0$ Hz, $\beta$-H), 3.84 (d, 2H, $J = 4.0$ Hz, $\beta$-H), 3.53 (s, 2H, $\beta$-H), 2.56–2.58 (d, 4H, $J = 4.5$ Hz, CH$_2$); [13]C NMR (126 MHz, CDCl$_3$): $\delta$ 165.87, 165.00, 156.88, 155.08, 150.58, 148.89, 144.98, 144.42, 142.65, 142.10, 141.20, 134.97, 134.05, 132.50, 130.37, 129.28, 128.66, 127.54, 127.36, 127.24, 126.55, 125.94, 124.63, 117.28, 116.96, 115.69, 111.47, 34.59; Ultraviolet-Vis-NIR(CH$_2$Cl$_2$): $\lambda_{max}$ ($\varepsilon$(M$^{-1}$ cm$^{-1}$)) 421 (12,000), 527 (5,500), 863 (2,200) nm; HR-MS (ESI-MS): $m/z = 1100.1529$, calcd for $(C_{64}H_{40}N_8Ni_2S_2)^+ = 1100.1519$ $[(M)^+]$.

**X-ray diffraction analysis.** X-ray diffraction data of **3b** were taken on a Bruker SMART APEX X-Ray diffractometer equipped with a large area CCD detector. X-ray diffraction data of **5b** were collected on CCD (MX225HE, Rayonix) with the synchrotron radiation ($\lambda = 0.8000$ Å) monochromated by the fixed exit Si (111)

double crystal at the BL38B1 in the SPring-8 with approval of the Japan Synchrotron Radiation Research Institute (JASRI) (proposal Nos. 2015B1397, 2016A1121). The oscillation angle, camera distance and exposure time per frame were 1°, 75 mm and 1 s, respectively. Two data sets consisting of 180 frames were integrated, scaled and merged with the programs HKL2000[27]. The structure was solved by SHELXT[28] and refined by least-squares calculations (SHELXL)[29] on $F^2$ for all reflections using the crystallographic software packages CrystalStructure[30]. All non-hydrogen atoms were refined with anisotropic displacement parameters and hydrogen atoms were placed in idealized positions and refined as rigid atoms with the relative isotropic displacement parameters. Crystallographic details are given in CIF files (Supplementary Data 1 and 2). The detailed crystallographic data for both the compounds are listed in Supplementary Table 1.

**HOMA calculations.** HOMA values of **3a**, **3b**, **5b** and Ni(II) porphyrin were calculated using C–C and C–N bond lengths of the X-ray crystallographic structure, according to the following equations:

$$HOMA = 1 - \alpha/n\Sigma\left(R_{opt} - R_i\right)^2 \quad (1)$$

where $n$ is the number of bonds taken into summation, $\alpha$ is an empirical constant, $R_{opt}$ is an optimal bond length and $R_i$ is a bond length of $i$th bond. $R_{opt} = 1.388$ (C–C) and 1.334 (C–N) Å and $\alpha = 257.7$ (C–C) and 93.52 (C–N) were used.

**TPA measurement.** The TPA spectrum was measured in the NIR region using the open-aperture $Z$-scan method with 130 fs pulses from an optical parametric amplifier (Light Conversion, TOPAS) operating at a repetition rate of 1 kHz generated from a Ti:sapphire regenerative amplifier system (Spectra-Physics, Hurricane). After passing through a 10 cm focal length lens, the laser beam was focused and passed through a 1 mm quartz cell. As the position of the sample cell could be controlled along the laser beam direction ($z$ axis) using the motor-controlled delay stage, the local power density within the sample cell could be simply controlled under constant laser intensity. The transmitted laser beam from the sample cell was then detected by the same photodiode as used for reference monitoring. The on-axis peak intensity of the incident pulses at the focal point, $I_0$, ranged from 40 to 60 GW cm$^{-2}$. For a Gaussian beam profile, the nonlinear absorption coefficient can be obtained by curve fitting of the observed open-aperture traces $T(z)$ with the following equation:

$$T(z) = 1 - \frac{\beta I_0\left(1 - e^{-\alpha_0 l}\right)}{2\alpha_0\left(1 + (z/z_0)^2\right)} \quad (2)$$

where $\alpha_0$ is the linear absorption coefficient, $l$ is the sample length and $z_0$ is the diffraction length of the incident beam. After the nonlinear absorption coefficient has been obtained, the TPA cross-section $\sigma_2$ of one solute molecule (in units of GM, where $1\,GM = 10^{-50}\,cm^4\,s\,photon^{-1}\,molecule^{-1}$) can be determined by using the following relationship:

$$\beta = \frac{\sigma_2 N_A C}{h\nu} \quad (3)$$

where $N_A$ is the Avogadro constant, $C$ is the concentration of the compound in solution, $h$ is the Planck constant and $\nu$ is the frequency of the incident laser beam.

**Theoretical calculations.** All calculations were carried out using the Gaussian 09 program[31]. Geometries of **3a** and **5b** for NICS and ACID calculations were obtained from their X-ray structures. All calculations were performed with Becke's three-parameter hybrid exchange functional and the Lee–Yang–Parr correlation functional (B3LYP)[32,33] and the 6-31G(d) basis set was used for all atoms. NBO calculations and Wiberg bond order analysis were performed with the PBE0 hybrid functional[34] and LANL2DZ combination of basis set and effective core potentials[35].

**Data availability.** Crystallographic data (CIF files) for **3b** and **5b** have been deposited with the Cambridge Crystallographic Data Centre as supplementary publications. CCDC 1484882 (**3b**) and CCDC 1484883 (**5b**) contain the supplementary crystallographic data. These data can be obtained free of charge from the Cambridge Crystallographic Data Centre via www.ccdc.cam.ac.uk/data_request/cif. All other data supporting the findings of this study are available within the article and its Supplementary Information.

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

## Acknowledgements

This work was supported by the Grant-in-Aid for Scientific Research on Innovative Areas (2601): π-System Figuration (JSPS KAKENHI grant numbers JP26102003 (to H.S.), JP15H00998 (to I.H.) and JP15H01001 (to S.S.) and by the Program for Leading Graduate Schools 'Integrative Graduate Education and Research in Green Natural Sciences' from MEXT, Japan. Crystallographic data were collected at the SPring-8 beam line BL38B1 with approval of the JASRI (proposal nos. 2015B1397 and 2016A1121). H.S. acknowledges the Asahi Glass Foundation for financial support. Research at Yonsei University was supported by the Samsung Science and Technology Foundation (SSTF-BA1402-10).

## Author contributions

H.S. designed and conducted the project and prepared the manuscript. R.N. and H.T. carried out the synthesis, characterization and determination of the optical properties. I.H. and J.-Y.S. carried out the X-ray diffraction analysis. W.-Y.C., Y.H. and D.K. measured and analysed the TPA properties. S.S. measured and analysed the MCD spectra. T.K. and S.I. conducted the theoretical analyses.

## Additional information

**Competing financial interests:** The authors declare no competing financial interests.

**Publisher's note**: 

