## [Peer Review File · Nature Communications]

Reviewer #1 (Remarks to the Author):

The hypothesis that stacking two planar antiaromatic π -systems together face-to-face should lead to a stable 3-dimensional dimer that exhibits characteristics of an aromatic system has roots in informal (unpublished) discussions that I remember from several decades ago. Eventually, Schleyer (ref. 8) and Fowler (ref. 9) published papers based on theory that unambiguously predicted this remarkable phenomenon. Antiaromatic systems generally exhibit exceptionally high reactivity (instability), whereas aromatic systems generally exhibit exceptionally low reactivity (stability). The predictions, therefore, constitute a molecular counterpart to the age-old adage “two wrongs make a right.”

Long before the Schleyer and Fowler publications (2007 and 2008, respectively), inspired by the early “folklore” on the subject, Christopher Adams attempted to synthesize a cyclophane with two antiaromatic cyclobutadienes held face-to-face (ref. 10). He never completed the difficult synthesis before he lost his position as an assistant professor, and that is the only attempt (that I know of) by anyone to verify experimentally this bold, long-standing hypothesis.

This brings us to the new work of Shinokubo *et al.* In my opinion, the experimental results describe in this manuscript provide the first definitive experimental proof that the “ π -stacked antiaromatic \Rightarrow aromatic” hypothesis is correct, and this makes the work worthy of publication in *Nature Communications*.

The evidence reported by Shinokubo *et al.* in support of this phenomenon is compelling and includes:

- 1) The spontaneous face-to-face aggregation of antiaromatic norcorrole **3b** in the solid state (Figure 2), at a distance closer than van der Waals repulsion would normally allow, which indicates intermolecular “bonding”.
- 2) Bond length “convergence” in the face-to-face trimer of antiaromatic norcorrole **3b** in the solid state. Antiaromatic compounds (e.g., cyclobutadiene) typically exhibit strong bond length alternation, to minimize electron delocalization, which is the cause of their antiaromaticity, whereas aromatic compounds typically exhibit strong bond convergence, to maximize electron delocalization, which is the source of their aromaticity (Supplementary Figure 3).
- 3) Large down-field shifts of the signals for the hydrogens on the rim of “intramolecular dimer” **5b** and the demonstration that shifting the equilibrium between the open and stacked conformations of **5b** toward more of the stacked form imparts larger down-field shifts (Supplementary Figures 19 & 20).
- 4) Bond length “convergence” in the face-to-face “intramolecular dimer” of antiaromatic norcorrole **5b** in the solid state (Supplementary Figure 3).
- 5) Remarkable stability of the face-to-face “intramolecular dimer” of antiaromatic norcorrole **5b**, compared to that of antiaromatic norcorrole **4b**, which does not form a dimer in solution (Figure 4).

The syntheses reported are elegant and efficient. The characterization of new compounds is thorough and sound. A detailed theoretical treatment of these systems is

entirely consistent with the observed experimental results and supports the conclusion that face-to-face aggregation of the antiaromatic norcorrole π -system produces a stable, 3-D structure that shows all the hallmarks of an aromatic π -system. The authors go on to show that the stacked antiaromatic π -systems exhibit unusual optical properties that should be tunable for possible future applications. This part is interesting, too, but the most important contribution, in my opinion, is the experimental validation of a theoretical concept that has stood untested for many decades.

Minor editorial suggestions:

Page 2, line 6 from the bottom: “experimental support to substantiate this claim” would be better as “experimental support to substantiate this **prediction**”

Page 2, last line: “contains two *meso*-carbons less” would be better as “contains two *meso*-carbons **fewer**”

Page 3, line 4: “adopts a triple-decker π – π stacked structure” would be better as “adopts a triple-decker π – π stacked structure **in the solid state**”

Page 3, line 9: “the first experimental piece of evidence” would be better as just “the first experimental evidence” (delete “piece of”)

Page 3, line 3 from the bottom: “spectrometry analyses” would be better as “**spectrometric** analyses”

Page 6, line 1: “conformers of **5b**, whereby the former predominates” would be better as “conformers of **5b**, **in which** the former predominates”

Figure 3a: The full name for reagent “DMI” should be spelled out in the caption, for non-experts. This same comment applies to the places in the Supp. Info. where DMI is mentioned.

Page 8, line 1: “broad absorption bands in the near IR region” should be expanded to be more explicit, e.g., “broad absorption bands in the near IR region ($\lambda > 1000$ nm)” or whatever region the authors wish to define as “the near IR region.”

Page 9 line 9: “It is feasible to assume” would be better as “It is **reasonable** to assume”

Page 10, second paragraph, lines 10-11: “between the lone pair orbitals on the carbon atoms of one monomer” would be better as “between the **occupied** orbitals on the carbon atoms of one monomer”

Page 10, second paragraph, last line: “previous theoretical findings for antiaromatic superphanes” would be better as “previous theoretical **predictions** for antiaromatic superphanes”

Table of Contents graphic: I recommend removing the question mark after the word “aromatic” in the Table of Contents graphic. Perhaps it should even be replaced with an exclamation point!

Reviewer #2 (Remarks to the Author):

3D-aromaticity is currently of great interest to researchers active in the porphyrinoid field (it was one of the main topics of interest at the recent ICCP-9 conference where the top porphyrin researchers meet) and in wider terms can be viewed as being on the cutting edge in theoretical terms where concept of bonding are concerned. This is a very important piece of work that clearly deserves to be published in Nature Comm. Having read the manuscript carefully the authors have carried out all the obvious measurements and calculations appropriate to the subject matter including the specialist MCD technique and I can see no obvious flaws in their methodology. The manuscript can be published essentially as is.

Reviewer #3 (Remarks to the Author):

Review of Paper 101565 from Shinokubo et al.

My comments are restricted to the crystal structure determinations of compounds 3b and 5b which are reported in the Supplementary Data for this paper and whose results are described in the MS.

Both samples have provided good data and reliable results. I would request attention only to some of the finer details in the results:

Compound 3b

In the text, Page S6, please check line 1 of the X-ray Diffraction Analysis section and remove the duplicated 'X-ray data were taken on a'.

Line 3: not 'was' but 'were'

Line 14: similarly, not 'has' but 'have'

Last line: not 'all compounds were' but 'both compounds are'.

Page S7, Figures: please enlarge these as far as allowed. In Figure 1, I would prefer to see the phenyl groups included, particularly in the left-hand view.

Page S8, Table 1, please check:

Empirical formula for 5b – include the solvent molecules, 2(CHCl₃)

The formula weights: I calculate 493.21 and 1341.37

For 3b, Z = 6

'obs' reflections should read 6291 and 9694 for 3b and 5b

'total reflections' should read 'No. of reflections measured'

Please include 'No. of unique reflections' with values of 8037 and 10347 for 3b and 5b, and 'Rint' with values of 0.036 and 0.077.

I suggest that all the comments in the CheckCIF report should be noted. Comments on the 'large residual density' in each structure would be appreciated.

It should also be noted that the molecule of 3b lies on a centre of symmetry and that two of the solvent molecules (CHCl₃) in 5b lie disordered about centres of symmetry. I wonder if the disorder in the C66 solvent molecules might give better geometry with C66 bound to Cl9, Cl7' and Cl8'?

In the main MS:

Page 4, line 5: please state clearly that Ni(2) of the central molecule of the triple-decker structure lies on a centre of symmetry. Then, on lines 9 and 10, 'As shown in Figure 2c,

the outer molecules are almost perfectly eclipsed...'

Figure 2: I would prefer to see the phenyl groups included in (b) and (c) – good clear views are feasible.

The words 'stack' and 'stacking' occur frequently in this paper, but refer only, I think, to triple-deck groups in 3b and paired rings in 5b. Is there any evidence of extended stacking in either of the crystalline samples or, indeed in solutions of these compounds?

Reviewer #4 (Remarks to the Author):

This is a highly interesting manuscript in which compelling and conclusive experimental evidence is provided for the occurrence of aromatic character in organic systems composed of anti-aromatic $4n$ pi-systems upon close face-to-face planar stacking with interplanar distances in the range of ca. 3.0 – 3.5 Ångström.

Hence, the authors provide the first experimental evidence for the ab initio theoretical prediction by Schleyer et al (ref. 8) that face-to-face stacking of anti-aromatic $4n$ pi-systems rings after incorporation in a superphane structure will result in through-space three-dimensional aromatic character on the basis of nucleus independent shift criteria (NICS).

This remarkable switch from anti-aromatic character into aromatic character was further substantiated by explicit ab initio calculation of the induced ring currents by Fowler et al. (ref. 9) on stacked anti-aromatic $4n$ pi-systems. Whereas the expected paratropic ring current was found for a single planar anti-aromatic $4n$ pi-system, upon planar face-to-face stacking the paratropic character of the single anti-aromatic $4n$ pi-system was quenched and replaced by layered diatropic currents in line with aromatic character (diatropic ring currents). Fowler et al provided also a general orbital model to rationalize these remarkable changes.

Thus, the close face-to-face stacking of anti-aromatic $4n$ pi-systems as such represents a novel approach to access aromatic systems next to triplet (cf. Wiberg, Chem. Rev., 2001, 101, 1317) and Mobius (cf. Rzepa, Chem. Rev., 2005, 105, 3697) strategies.

The present authors use a formally anti-aromatic 16-pi electron ring-contracted porphyrin norcorrole Ni(II) complex, which is synthetically accessible on a gram scale (ref. 11). For the current study the sterically bulky mesityl side-groups (3a) were replaced by the smaller phenyl groups (3b).

For clarity for the non-expert I advise to high-light the 16-pi electron perimeter of the ring-contracted norcorrole Ni(II) complex (for example in structures 3a/3b (Fig. 1)).

Whereas in solution (^1H NMR) 3b the pyrrole protons are positioned upfield at 1.7-2.2 ppm in line with the anti-aromatic character of the solvated norcolle Ni(II) complex, upon crystallization a triple-decker stacking structure is found in contrast to the herringbone packing structure found for less bulky 3a (ref. 11).

Single crystal X-ray structure analysis of the triple-decker structure revealed that bond length alternation (BLA) for 3b is significantly smaller than in the case of 3a (ref. 11). I agree that this is compelling evidence (according to the harmonic oscillator model of aromaticity (HOMA)) that upon face-to-face stacking in the solid state 3b apparently appears to undergo some change from anti-aromatic into aromatic due to face-to-face stacking.

This is further substantiated by the properties of the synthesized tethered dimers 5a (with mesityl side groups) and 5b (with phenyl side groups). Only the latter (5b) gave a face-to-face stacked single crystal X-ray structure. Similar BLA changes were found for 5b as for 3b in the solid state. Rewardingly for 5b ¹H NMR the pyrrole protons are found 3.5-4.7 ppm and shift even further downfield upon lowering the temperature. This is indeed indicative for an equilibrium shift between non-stacked and stacked conformers; a similar temperature dependence is not observed in the case of 5a.

The authors have further done ab initio anisotropy of the induced current (ACID) calculations (Fig. 3). However, the ACID plots are too small in size to establish visually that an attenuated current density is found in the case of 5b. The authors have to improve the quality (size!) of Fig. 3c or transfer the ACID plots to the Supporting Information. In the case of ACID plots it is important that the number and direction of the arrows is discernable.

Similarly the two-dimensional NICS plot (Supporting Information Fig. 6) is also not clear. Due to its size the two-dimensional NICS plot is difficult to interpret. It would be nice when some numerical NICS values were added to the manuscript showing the difference in magnetic response between the single molecules and after face-to-face stacking (for example in the case of 5a and 5b).

Another interesting experiment is the comparison of the chemical stability of 4b and 5b. I agree with the authors that face-to-face stacking (5b) indeed has a marked effect on the stability of the face-to-face stacked tethered dimer. This indicates that 5b has become more aromatic.

Further support for the change of anti-aromatic character in the case of 4b in going to the face-to-face stacked aromatic dimer 5b is given by a comparison of their optical data. I agree with the authors that the optical data support that the electronic structure of 5b differs from that of the monomer. This is substantiated by TD-DFT calculations and the magnetic circular dichroism (MCD) results. In the case of 5b the transition at 850 nm appears to be derived from a transition between the face-to-face stacked norcorrole units.

Finally, the authors have studied the two-photon absorption (TPA) properties of their systems. The face-to-face stacked dimer 5b indeed possesses a considerably enhanced TPA cross-section value than the single molecules. Again this is in line with a conversion of anti-aromatic character in single molecules into aromatic character upon face-to-face stacking.

In my opinion this manuscript is suitable for publication in Nature Communications. Notwithstanding, some modifications are required.

The authors provide the first proper experimental example that upon face-to-face stacking anti-aromatic 4n pi-systems are converted become 3D-aromatic. The authors are to be complemented with their thorough experimental work!

Comments form Reviewer #1

The hypothesis that stacking two planar antiaromatic π -systems together face-to-face should lead to a stable 3-dimensional dimer that exhibits characteristics of an aromatic system has roots in informal (unpublished) discussions that I remember from several decades ago. Eventually, Schleyer (ref. 8) and Fowler (ref. 9) published papers based on theory that unambiguously predicted this remarkable phenomenon. Antiaromatic systems generally exhibit exceptionally high reactivity (instability), whereas aromatic systems generally exhibit exceptionally low reactivity (stability). The predictions, therefore, constitute a molecular counterpart to the age-old adage “two wrongs make a right.”

Long before the Schleyer and Fowler publications (2007 and 2008, respectively), inspired by the early “folklore” on the subject, Christopher Adams attempted to synthesize a cyclophane with two antiaromatic cyclobutadienes held face-to-face (ref. 10). He never completed the difficult synthesis before he lost his position as an assistant professor, and that is the only attempt (that I know of) by anyone to verify experimentally this bold, long-standing hypothesis.

This brings us to the new work of Shinokubo *et al.* In my opinion, the experimental results describe in this manuscript provide the first definitive experimental proof that the “ π -stacked antiaromatic \Rightarrow aromatic” hypothesis is correct, and this makes the work worthy of publication in *Nature Communications*.

The evidence reported by Shinokubo *et al.* in support of this phenomenon is compelling and includes:

1) The spontaneous face-to-face aggregation of antiaromatic norcorrole **3b** in the solid state (Figure 2), at a distance closer than van der Waals repulsion would normally allow, which indicates intermolecular “bonding”.

2) Bond length “convergence” in the face-to-face trimer of antiaromatic norcorrole **3b** in the solid state. Antiaromatic compounds (e.g., cyclobutadiene) typically exhibit strong bond length alternation, to minimize electron delocalization, which is the cause of their antiaromaticity, whereas aromatic compounds typically exhibit strong bond convergence, to maximize electron delocalization, which is the source of their aromaticity (Supplementary Figure 3).

3) Large down-field shifts of the signals for the hydrogens on the rim of “intramolecular dimer” **5b** and the demonstration that shifting the equilibrium between the open and stacked conformations of **5b** toward more of the stacked form imparts larger down- field shifts (Supplementary Figures 19 & 20).

4) Bond length “convergence” in the face-to-face “intramolecular dimer” of antiaromatic norcorrole **5b** in the solid state (Supplementary Figure 3).

5) Remarkable stability of the face-to-face “intramolecular dimer” of antiaromatic norcorrole **5b**, compared to that of antiaromatic norcorrole **4b**, which does not form a dimer in solution (Figure 4).

The syntheses reported are elegant and efficient. The characterization of new compounds is thorough and sound. A detailed theoretical treatment of these systems is entirely consistent with the observed experimental results and supports the conclusion that face-to-face aggregation of the antiaromatic norcorrole π -system produces a stable, 3-D structure that shows all the hallmarks of an aromatic π -system. The authors go on to show that the stacked antiaromatic π -systems exhibit unusual optical properties that should be tunable for possible future applications. This part is interesting, too, but the most important contribution, in my opinion, is the experimental validation of a theoretical concept that has stood untested for many decades.

Minor editorial suggestions:

Page 2, line 6 from the bottom: “experimental support to substantiate this claim” would be better as “experimental support to substantiate this **prediction**”

Page 2, last line: “contains two *meso*-carbons less” would be better as “contains two *meso*-carbons **fewer**”

Page 3, line 4: “adopts a triple-decker π - π stacked structure” would be better as “adopts a triple-decker π - π stacked structure **in the solid state**”

Page 3, line 9: “the first experimental piece of evidence” would be better as just “the first experimental evidence” (delete “piece of”)

Page 3, line 3 from the bottom: “spectrometry analyses” would be better as “**spectrometric** analyses”

Page 6, line 1: “conformers of **5b**, whereby the former predominates” would be better as “conformers of **5b**, **in which** the former predominates”

Figure 3a: The full name for reagent “DMI” should be spelled out in the caption, for non- experts. This same comment applies to the places in the Supp. Info. where DMI is mentioned.

Page 8, line 1: “broad absorption bands in the near IR region” should be expanded to be more explicit, e.g., “broad absorption bands in the near IR region ($\lambda > 1000$ nm)” or whatever region the authors wish to define as “the near IR region.”

Page 9 line 9: “It is feasible to assume” would be better as “It is **reasonable** to assume”

Page 10, second paragraph, lines 10-11: “between the lone pair orbitals on the carbon atoms of one monomer” would be better as “between the **occupied** orbitals on the carbon atoms of one monomer”

Page 10, second paragraph, last line: “previous theoretical findings for antiaromatic superphanes” would be better as “previous theoretical **predictions** for antiaromatic superphanes”

Table of Contents graphic: I recommend removing the question mark after the word 'aromatic' in the Table of Contents graphic. Perhaps it should even be replaced with an exclamation point!

Our responses

We are so delighted at these supportive and encouraging comments. We have revised our manuscript according to his editorial suggestions.

Comments form Reviewer #2

3D-aromaticity is currently of great interest to researchers active in the porphyrinoid field (it was one of the main topics of interest at the recent ICPP-9 conference where the top porphyrin researchers meet) and in wider terms can be viewed as being on the cutting edge in theoretical terms where concept of bonding are concerned. This is a very important piece of work that clearly deserves to be published in Nature Comm. Having read the manuscript carefully the authors have carried out all the obvious measurements and calculations appropriate to the subject matter including the specialist MCD technique and I can see no obvious flaws in their methodology. The manuscript can be published essentially as is.

Our responses

We are so delighted at these supportive comments.

Comments form Reviewer #3

My comments are restricted to the crystal structure determinations of compounds **3b** and **5b** which are reported in the Supplementary Data for this paper and whose results are described in the MS.

Both samples have provided good data and reliable results. I would request attention only to some of the finer details in the results:

Compound 3b

In the text, Page S6, please check line 1 of the X-ray Diffraction Analysis section and remove the duplicated ‘X-ray data were taken on a’.

Line 3: not ‘was’ but ‘were’

Line 14: similarly, not ‘has’ but ‘have’

Last line: not ‘all compounds were’ but ‘both compounds are’.

Our responses

We have corrected these typos as suggested.

Page S7, Figures: please enlarge these as far as allowed. In Figure 1, I would prefer to see the phenyl groups included, particularly in the left-hand view.

Our responses

We have enlarged the size of these figures. We have revised Figure 1 to include the phenyl groups as suggested.

Page S8, Table 1, please check:

Empirical formula for **5b** – include the solvent molecules, 2(CHCl₃)

The formula weights: I calculate 493.21 and 1341.37

For **3b**, Z = 6

‘obs’ reflections should read 6291 and 9694 for **3b** and **5b**

‘total reflections’ should read ‘No. of reflections measured’

Please include ‘No. of unique reflections’ with values of 8037 and 10347 for **3b** and **5b**, and ‘Rint’ with values of 0.036 and 0.077.

Our responses

We have corrected the data in Table S1 as suggested.

I suggest that all the comments in the CheckCIF report should be noted. Comments on the ‘large residual density’ in each structure would be appreciated.

Our responses

We have included the following comments on the ‘large residual density’ in the revised cif files.

“When observed closely, the electron density appears to be near to the metal centres. Thus, such residual density is likely not from unaccounted atom types.”

It should also be noted that the molecule of **3b** lies on a centre of symmetry and that two of the solvent molecules (CHCl₃) in **5b** lie disordered about centres of symmetry. I wonder if the disorder in the C66 solvent molecules might give better geometry with C66 bound to C19, C17' and C18'?

Our responses

Thank you for this comment on geometry of the chloroform molecule (C66). According to the reviewer's suggestion, we attempted to re-build a chloroform molecule with C66, C19, C17', and C18' atoms. However, after careful refinement of the structure, we concluded that geometry of the original chloroform molecule is better than that of the re-built one. Therefore, we decided to adopt our original structure.

In the main MS:

Page 4, line 5: please state clearly that Ni(2) of the central molecule of the triple-decker structure lies on a centre of symmetry. Then, on lines 9 and 10, 'As shown in Figure 2c, the outer molecules are almost perfectly eclipsed...'

Our responses

We have stated that Ni(2) of the central molecule is located on the centre of symmetry as suggested.

Figure 2: I would prefer to see the phenyl groups included in (b) and (c) – good clear views are feasible.

Our responses

We have included the phenyl groups in Figures 2b and 2c as suggested.

The words 'stack' and 'stacking' occur frequently in this paper, but refer only, I think, to triple-deck groups in **3b** and paired rings in **5b**. Is there any evidence of extended stacking in either of the crystalline samples or, indeed in solutions of these compounds?

Our responses

We have not seen any extended stacking rather than triple and double stacking structures in **3b** and **5b**. We do hope to see such an infinite stacking of antiaromatic porphyrins in the future.

Comments form Reviewer #4

This is a highly interesting manuscript in which compelling and conclusive experimental evidence is provided for the occurrence of aromatic character in organic systems composed of anti-aromatic $4n$ pi-systems upon close face-to-face planar stacking with interplanar distances in the range of ca. 3.0 – 3.5 Ångström.

Hence, the authors provide the first experimental evidence for the ab initio theoretical prediction by Schleyer et al (ref. 8) that face-to-face stacking of anti-aromatic $4n$ pi-systems rings after incorporation in a superphane structure will result in through-space three-dimensional aromatic character on the basis of nucleus independent shift criteria (NICS).

This remarkable switch from anti-aromatic character into aromatic character was further substantiated by explicit ab initio calculation of the induced ring currents by Fowler et al. (ref. 9) on stacked anti-aromatic $4n$ pi-systems. Whereas the expected paratropic ring current was found for a single planar anti-aromatic $4n$ pi-system, upon planar face-to-face stacking the paratropic character of the single anti-aromatic $4n$ pi-system was quenched and replaced by layered diatropic currents in line with aromatic character (diatropic ring currents). Fowler et al provided also a general orbital model to rationalize these remarkable changes.

Thus, the close face-to-face stacking of anti-aromatic $4n$ pi-systems as such represents a novel approach to access aromatic systems next to triplet (cf. Wiberg, Chem. Rev., 2001, 101, 1317) and Mobius (cf. Rzepa, Chem. Rev., 2005, 105, 3697) strategies.

The present authors use a formally anti-aromatic 16-pi electron ring-contracted porphyrin norcorrole Ni(II) complex, which is synthetically accessible on a gram scale (ref. 11). For the current study the sterically bulky mesityl side-groups (**3a**) were replaced by the smaller phenyl groups (**3b**).

Our responses

We highly appreciate for these supportive comments.

For clarity for the non-expert I advise to high-light the 16-pi electron perimeter of the ring-contracted norcorrole Ni(II) complex (for example in structures **3a/3b** (Fig. 1)).

Our responses

We have high-lighted the conjugation circuit of 16-pi norcorroles with bold lines in Figure 1 as suggested.

Whereas in solution (^1H NMR) **3b** the pyrrole protons are positioned upfield at 1.7-2.2 ppm in line with the anti-aromatic character of the solvated norcorrole Ni(II) complex, upon crystallization a triple-decker stacking structure is found in contrast to the herringbone packing structure found for less bulky **3a** (ref. 11).

Single crystal X-ray structure analysis of the triple-decker structure revealed that bond length alternation (BLA) for **3b** is significantly smaller than in the case of **3a** (ref. 11). I agree that this is compelling evidence (according to the harmonic oscillator model of aromaticity (HOMA)) that upon face-to-face stacking in the solid state **3b** apparently appears to undergo some change from anti-aromatic into aromatic due to face-to-face stacking.

This is further substantiated by the properties of the synthesized tethered dimers **5a** (with mesityl side groups) and **5b** (with phenyl side groups). Only the latter (**5b**) gave a face-to-face stacked single crystal X-ray structure. Similar BLA changes were found for **5b** as for **3b** in the solid state. Rewardingly for **5b** ^1H NMR the pyrrole protons are found 3.5-4.7 ppm and shift even further downfield upon lowering the temperature.

This is indeed indicative for an equilibrium shift between non-stacked and stacked conformers; a similar temperature dependence is not observed in the case of **5a**.

The authors have further done ab initio anisotropy of the induced current (ACID) calculations (Fig. 3). However, the ACID plots are too small in size to establish visually that an attenuated current density is found in the case of **5b**. The authors have to improve the quality (size!) of Fig. 3c or transfer the ACID plots to the Supporting Information. In the case of ACID plots it is important that the number and direction of the arrows is discernable.

Our responses

We appreciate for this helpful advice. We have moved the ACID plots to the Supporting Information and enlarged the size as large as possible (Supporting Figure 18).

Similarly the two-dimensional NICS plot (Supporting Information Fig. 6) is also not clear. Due to its size the two-dimensional NICS plot is difficult to interpret. It would be nice when some numerical NICS values were added to the manuscript showing the difference in magnetic response between the single molecules and after face-to-face stacking (for example in the case of **5a** and **5b**).

Our responses

We have enlarged the size of the two-dimensional NICS plots as large as possible (Supporting Figure 19). Furthermore, we have provided some numerical NICS values in Figure 3c as suggested.

Another interesting experiment is the comparison of the chemical stability of **4b** and **5b**. I agree with the authors that face-to-face stacking (**5b**) indeed has a marked effect on the stability of the face-to-face stacked tethered dimer. This indicates that **5b** has become more aromatic.

Further support for the change of anti-aromatic character in the case of **4b** in going to the face-to-face stacked aromatic dimer **5b** is given by a comparison of their optical data. I agree with the authors that the optical data support that the electronic structure of **5b** differs from that of the monomer. This is substantiated by TD-DFT calculations and the magnetic circular dichroism (MCD) results. In the case of **5b** the transition at 850 nm appears to be derived from a transition between the face-to-face stacked norcorrole units.

Finally, the authors have studied the two-photon absorption (TPA) properties of their systems. The face-to-face stacked dimer **5b** indeed possesses a considerably enhanced TPA cross-section value than the single molecules. Again this is in line with a conversion of anti-aromatic character in single molecules into aromatic character upon face-to-face stacking.

In my opinion this manuscript is suitable for publication in Nature Communications. Notwithstanding, some modifications are required. The authors provide the first proper experimental example that upon face-to-face stacking anti-aromatic $4n$ pi-systems are converted become 3D-aromatic. The authors are to be complemented with their thorough experimental work!

Our responses

It is our great pleasure to hear these supportive comments.